# Extracellular Vesicles Act as Carriers for Cargo Delivery and Regulate Wnt Signaling in the Hepatocellular Carcinoma Tumor Microenvironment

**DOI:** 10.3390/cancers15072088

**Published:** 2023-03-31

**Authors:** Risheng He, Yi Xu, Liang Yu, Nanfeng Meng, Hang Wang, Yunfu Cui, Judy Wai Ping Yam

**Affiliations:** 1Department of Hepatopancreatobiliary Surgery, The Second Affiliated Hospital of Harbin Medical University, Harbin 150086, China; 2Key Laboratory of Basic Pharmacology of Ministry of Education, Zunyi Medical University, Zunyi 563006, China; 3Key Laboratory of Functional and Clinical Translational Medicine, Fujian Province University, Xiamen Medical College, Xiamen 361000, China; 4Jiangsu Province Engineering Research Center of Tumor Targeted Nano Diagnostic and Therapeutic Materials, Yancheng Teachers University, Yancheng 224007, China; 5Key Laboratory of Biomarkers and In Vitro Diagnosis Translation of Zhejiang Province, Hangzhou 310063, China; 6Key Laboratory of Gastrointestinal Cancer, Ministry of Education, School of Basic Medical Sciences, Fujian Medical University, Fuzhou 350122, China; 7State Key Laboratory of Chemical Oncogenomics, Key Laboratory of Chemical Genomics, Peking University Shenzhen Graduate School, Shenzhen 518055, China; 8Key Laboratory of Intelligent Pharmacy and Individualized Therapy of Huzhou, Department of Pharmacy, Changxing People’s Hospital, Changxing 313000, China; 9Department of Pathology, Li Ka Shing Faculty of Medicine, The University of Hong Kong, Hong Kong SAR 999077, China

**Keywords:** hepatocellular carcinoma, extracellular vesicles, Wnt signaling

## Abstract

**Simple Summary:**

As one of the most malignant cancers, the mechanisms underlying the occurrence and development of hepatocellular carcinoma (HCC) are complex, and the development of effective treatment strategies has been difficult. Extracellular vesicles (EVs) participate in HCC development in various ways. This review summarizes the latest research on EVs related to Wnt signaling in HCC and expounds on the underlying mechanisms, providing a reference for novel HCC treatment strategies.

**Abstract:**

As the primary type of liver cancer, hepatocellular carcinoma (HCC) causes a large number of deaths every year. Despite extensive research conducted on this disease, the prognosis of HCC remains unclear. Recently, research has largely focused on extracellular vesicles (EVs), and they have been found to participate in various ways in the development of various diseases, including HCC, such as by regulating cell signaling pathways. However, recent studies have reported the mechanisms underlying the regulation of Wnt signaling by EVs in HCC, primarily focusing on the regulation of the canonical pathways. This review summarizes the current literature on the regulation of Wnt signaling by EVs in HCC and their underlying mechanisms. In addition, we also present future research directions in this field. This will deepen the understanding of HCC and provide new ideas for its treatment.

## 1. Introduction

### 1.1. Hepatocellular Carcinoma (HCC)

HCC is the most common type of liver cancer and progresses as one of the deadliest malignancies, accounting for approximately 90% of primary liver cancers. Furthermore, men are more prone to this disease than women. The incidence of HCC varies in different regions, and Asia has been reported to have a high incidence of HCC. Its occurrence is primarily attributed to factors such as viral infections, toxins, immune system diseases, metabolic diseases as well as alcohol consumption. Moreover, HCC causes more than 700,000 deaths per year worldwide [1,2]. The current preferred treatment method entails early diagnosis and surgical resection of the tumor. However, because the early symptoms of HCC are insignificant, a large number of patients are diagnosed at the advanced stage of the disease. Therefore, the optimal treatment time has passed. In such cases, treatment options are limited and mainly include local treatments, such as radiofrequency ablation (RFA) and transarterial chemoembolization (TACE), chemotherapy, targeted therapy, and immunotherapy. However, the treatment is usually unsatisfactory, largely affecting the survival of patients [3]. According to a recent cancer statistics report, the survival rate of patients with common cancers, including HCC, has significantly improved to some extent. However, HCC prognosis continues to remain unsatisfactory, with a 5-year survival rate of only 18% due to its high metastatic capacity and recurrence rate [4]. HCC is currently the third leading cause of cancer-related deaths around the world [5]. Therefore, summarizing the underlying mechanism of its occurrence and progression will help us better understand the disease and provide a reference for subsequent research.

### 1.2. Extracellular Vesicles (EVs) and Tumor Microenvironment (TME)

In the 1960s, Wolf discovered a substance that had never been reported before, which he named “platelet dust” because it was isolated from platelets. This was the first report on EVs. Furthermore, Wolf discovered that they had sudanophilic properties, indicating they were rich in lipids [6]. Using related technology, it has been reported that EVs have a phospholipid bilayer structure, and these lipid-membrane-coated vesicles are present in almost all human cell types. They are especially abundant in body fluids such as blood, saliva, and urine [7,8]. Although the classification of EVs is constantly evolving, they are generally categorized as ectosomes and exosomes. Vesicles that are generated by the direct outward budding of the plasma membrane are called ectosomes. The budding of the plasma membrane produces microvesicles, microparticles, and large vesicles that are approximately 50 nm–1 μm in diameter [9]. In contrast, exosomes are of endosomal origin and are approximately 40–160 nm in diameter (approximately 100 nm on average) [10,11]. In addition to the abovementioned types, the EVs also include apoptotic bodies (100–5000 nm in diameter) and arrestin domain-containing protein 1-mediated microvesicles (approximately 50 nm in diameter) [12,13]. At the time of the initial discovery, EVs were identified as a vehicle for the cell to eliminate waste compounds [14]. Research progression has made it increasingly clear that EVs serve various functions apart from being waste carriers. They exchange cargoes, such as nucleic acids, lipids, and proteins, between cells through complex mechanisms, and the main processes include the docking of EVs with the plasma membrane, activation of surface receptors and signals, and the internalization of EVs by recipient cells [15]. Thus, EVs regulate various physiological and pathological processes [16,17]. Moreover, increasing evidence suggests that EVs are not only involved in the occurrence and development of multiple tumor types, including gastrointestinal tumors but also harbor diagnostic and therapeutic potential [18,19,20]. For example, Urban et al. reported that EV phenotyping, especially in combination with serum alpha-fetoprotein, represents a minimally invasive and accurate tool for a liquid biopsy that could improve cancer screening and the differential diagnosis of hepatobiliary malignancies [21].

Tumor cells, along with their extracellular environment, constitute an extraordinarily complex system known as the TME, which also contains various stromal cells and noncellular components apart from tumor cells. Most stromal cells consist of cancer-associated fibroblasts (CAFs), endothelial cells, immune and inflammatory cells such as tumor-associated macrophages (TAMs), regulatory T cells, and dendritic cells, whereas the noncellular components include the extracellular matrix, cytokines, and growth factors [22]. Tumor progression requires bidirectional communication between tumor cells and their surrounding environments, and EVs are key elements in this process [23,24]. In a tumor microenvironment, EVs carry cargoes to initiate or facilitate cancer progression at various stages, such as during the proliferation of cells and escape from apoptosis, angiogenesis, cell invasion, and metastasis, reprogramming energy metabolism, evasion of the immune response, and mutation transfer [25]. For instance, tumor cell-derived exosomes are rich in the programmed cell death ligand 1 (PD-L1), an immunosuppressive signaling ligand that binds to PD-1 on the surface of CD8+ T cells, reducing the production and secretion of cytokines by T cells and preventing the immune cells from killing tumors. This leads to the further transmission of immunosuppressive signals and eliminates the direct contact between the tumor and immune cells [26]. In contrast, nature killer (NK) cells are an important type of innate immune cells and play a key role in immune surveillance. NK cell-derived exosomes are rich in killer proteins that establish direct contact with the tumor site and exert antitumor functions to overcome the drawback of the requirement to reach the tumor site before they can function [27].

### 1.3. Wnt Signaling

Wnt signaling was recognized and unraveled in 1973 in a study on Drosophila melanogaster. At that time, scientists discovered a wingless (Wg) gene during the screening for visual phenotypic mutations [28]. During related research, this gene and the intracellular signaling pathway that is regulated by it were found to be closely linked to the embryonic development of Drosophila melanogaster [29,30]. This signaling pathway was found to be linked to tumors when the activation of the Int-1 gene in mouse breast tumors was discovered. Studies reported that the gene was activated in two ways: insertion of the provirus into the genetic locus and transgenic overexpression [31,32,33]. Int-1 was later designated as Wnt1 and was reported to be a homolog of Wg [34]. Intriguingly, axis duplication was induced when the murine Wnt1 mRNA was injected into Xenopus embryos [35]. These observations support the notion that Wnt homologs and Wnt signaling are highly conserved and may play important roles in the normal formation and development of tumors. Wnt signaling was first reported in human disease when familial adenomatous polyposis was caused by mutations in the adenomatous polyposis coli (APC) gene to some extent and eventually evolved into colon cancers [36,37]. Two years after the publication of these reports, APC was discovered to function by interacting with β-catenin [38,39]. Loss of APC activity contributed to the activation of T-cell factor (TCF)4/β-catenin signaling, and TCF was identified as a nuclear effector of Wnt signaling [40]. Numerous components of Wnt signaling have been discovered in the last three decades, and these components have been reported to be closely related to various physiological and pathological processes [41,42]. Wnt signaling functions mainly via three primary intracellular signaling cascades. Apart from the canonical pathway of Wnt/β-catenin signaling, two noncanonical pathways comprise Wnt/planar cell polarity (Wnt/PCP) signaling and Wnt/Ca^2+^ signaling (Figure 1).

#### 1.3.1. Canonical Pathway

The canonical pathway consists of three primary components: Wnt-associated membrane proteins; degradation complexes that decide the fate of β-catenin; and key transcriptional regulators in the nucleus. Among these, some compounds have a cancer-promoting function, and others act as tumor suppressors. Mutagenesis or loss of function of these components is critically important for Wnt/β-catenin signaling and gene expression. To date, 19 different Wnt ligands have been reported. They are secreted glycoproteins that are rich in cysteine [43,44]. Of these, eight have been reported to be involved in the canonical pathway, such as Wnt1 and Wnt2 [45].

Activation of the Wnt/β-catenin signaling requires the combination of Wnt ligands with the seven-pass transmembrane frizzled (FZD) that is localized adjacent to the low-density lipoprotein receptor-related protein (LRP), a coreceptor of FZD [46]. The LRP family encompasses multiple isoforms, and LRP5/6 has certain functions in vertebrates; however, these are different from Arrow in Drosophila [41]. Subsequently, the receptor–ligand complex transduces the initiation signal into intracellular responses, and the degradation complexes are mobilized. Current research reveals that the degradation complexes primarily comprise the axis inhibition protein (AXIN), disheveled (DVL), glycogen synthase kinase 3β (GSK3β), APC, WTX, casein kinase 1α (CK1α), β-transducin repeats-containing proteins (β-TrCP), yes-associated protein (YAP), and PDZ-binding motif (TAZ) [47,48,49]. They are responsible for regulating the stability of β-catenin. When Wnt signaling is activated, the intracellular LPR region containing PPPSP phosphorylation motifs is phosphorylated by CK1γ, GSK3β, and Cyclin Y. Thus, presenting remarkable opportunities for the combination of LPR with AXIN [50,51]. Simultaneously, DVL moves toward the cell membrane, leading to the accumulation of DVL in the cell membrane. Various protein kinases concomitantly induce DVL phosphorylation and then interact with FZD, exposing the binding sites for AXIN [42]. The binding of AXIN to the two abovementioned proteins results in functional limitations of degradation complexes, which are partially due to the inhibition of GSK3 activities. All these processes contribute to the stabilization and accumulation of β-catenin, and finally, they are transferred to the nucleus. In the nucleus, β-catenin substitutes for transducing-like enhancer protein (TLE)/Groucho and binds to TCF and lymphoid enhancer factor (LEF). Moreover, it recruits several histone-modifying co-activators that eventually initiate transcription [52,53].

In the absence of Wnt ligand binding to receptors, the pathway exists in an inactive state which has been attributed to low levels of β-catenin. The degradation is primarily induced via the following mechanisms. AXIN is a scaffold protein that is a part of degradation complexes and recruits other components [54]. AXIN is phosphorylated by GSK3β when the pathway is inactive, increasing its affinity to β-catenin [55,56]. Simultaneously, GSK3β phosphorylates β-catenin, and this process is assisted by AXIN and APC [57]. β-catenin is also phosphorylated by CK1α [58]. Furthermore, YAP/TAZ becomes a temporary part of the degradation complexes and recruits β-TrCP [49]. Subsequently, β-catenin detaches from the degradation complexes and is ubiquitinated by b-TrCP, leading to β-catenin degradation by the proteasome [59]. The absence of β-catenin in the nucleus results in the binding of TCF/LEF to TLE/Groucho, and this repressive complex recruits histone deacetylase (HDAC), repressing gene transcription [60].

#### 1.3.2. Noncanonical Pathways

Noncanonical pathways are also initiated by the combination of Wnt ligands with FZD. Simultaneously, the ligand also binds to tyrosine kinase co-receptors, such as the RAR-related orphan receptor (ROR) [61]. When the signal of this interaction is transmitted to the cytoplasm during Wnt/PCP signaling, DVL is rapidly mobilized and mediates the activation of the Ras homolog family member A transforming protein (RHOA) and Ras-related C3 botulinum toxin substrate 1 (RAC1) [62]. The former further activates its downstream protein RHO-associated kinase (ROCK), resulting in the modification of cytoskeletal rearrangement and actin cytoskeleton [45]. The latter activates JUN N-terminal kinase (JNK) via phosphorylation and transmits signals to the nucleus, initiating the transcription of the related genes [63]. During Wnt/Ca^2+^ signaling, Wnt ligands, in combination with FZD, activate phospholipase C (PLC), which upregulates the intracellular Ca^2+^ levels via the hydrolysis of phosphatidylinositol (4,5)-biphosphates (PIP2) to diacylglycerol (DAG) and inositol (1,4,5)-triphosphates (IP3). Previous studies have reported that DAG activates protein kinase C (PKC). Simultaneously, the induction of IP3 leads to the release of Ca^2+^ ions that are intracellularly stored. Furthermore, high levels of Ca^2+^ activate Ca^2+^/calmodulin-dependent kinase II (CaMKII) as well as calcineurin (CaN). The former induces the activation of Nemo-like kinase (NLK) by phosphorylating TGFβ-activated kinase 1 (TAK1), which suppresses the Wnt/β-catenin signaling. The latter leads to the activation of the nuclear factor of activated T-cells (NFAT) translocating to the nucleus, where it regulates relevant target gene expression. This pathway has been reported to be closely associated with cytoskeleton organization and cell motility [64,65]. 

## 2. EVs Regulate Wnt Signaling in HCC

So far, EVs have played an important role in the progression of multiple cancers by regulating the Wnt pathway. The migration of cancer cells in breast cancer is closely related to stromal mobilization, and studies have reported that this can be achieved by Wnt/PCP signaling regulation by EVs [66]. In cervical cancer-derived EVs, Wnt7b mRNA is expressed at high levels. It could be transferred to human umbilical vein endothelial cells to promote Wnt7b synthesis in recipient cells, affecting the proliferation and angiogenesis of recipient cells by regulating the β-catenin signaling, ultimately promoting tumor progression [67]. However, the regulation of Wnt signaling by EVs in HCC is beginning to be researched. Current studies have found that Evs can transfer contents between different HCC cells and activate Wnt signaling, thereby regulating the biological behavior of recipient cells and promoting the progression of malignant phenotypes. In addition, stromal cell-derived Evs can also deliver contents to HCC cells and other stromal cell types. These, on the one hand, lead to HCC resistance and, on the other hand, lead to immune cells expressing more exhaustion markers and fewer effector molecules. This review summarizes the current status and mechanisms underlying EV regulation of Wnt signaling in HCC to help us gain a better understanding of this phenomenon and provide ideas for deriving new therapeutic strategies for HCC.

### 2.1. Evs Regulate Wnt Signaling via Transport between Different Types of HCC Cells

The regulation of Wnt signaling by Evs in HCC is primarily achieved by modulating the classical pathway. Several studies have found that Evs regulate Wnt signaling via transport between different types of HCC cells (Table 1). Solid tumor cells are often surrounded by a hypoxic microenvironment [68], a condition favoring metastatic tumor progression [69]. Yu et al. discovered that this hypoxic environment contributes to exosome production in HCC. Furthermore, exosomes secreted by hypoxic tumor cells were found to promote malignant phenotypes of normoxic cells, including their proliferation, migration, invasion, and epithelial-to-mesenchymal transition (EMT). A study of the underlying mechanisms found that miR-1273f levels increased significantly in exosomes secreted by Huh7 cells under a hypoxic environment, and recipient HCC cells showed enhanced malignancy that was closely related to this miRNA in vitro. The search for downstream targets revealed that miR-1273f functions by targeting a tumor suppressor gene (LHX6) [70]. Several previous studies have demonstrated that LHX6 can inhibit the progression of multiple tumor types, including glioma, lung cancer, and breast cancer. LHX6 regulates Wnt/β-catenin signaling by inhibiting β-catenin expression, thereby inhibiting tumor progression. Furthermore, this process can be reversed by the downregulation of LHX6 by miR-1273f [71,72,73,74].

Accumulating evidence suggests that via exosome-mediated material transfer, high-metastatic HCC cells can enhance the migration and invasion abilities of low-metastatic HCC cells [75,76,77]. An in vitro study that isolated exosomes from highly metastatic HCC cells and used them to treat low-metastatic HCC cells observed reduced apoptosis and enhanced proliferative capacity in the latter. The abilities of the recipient cells to invade and migrate were also simultaneously enhanced. Animal experiments concluded that the tumorigenic rate of mice injected with exosome-treated tumor cells increased significantly, and multiple metastases were found in the liver and lungs, suggesting enhanced tumor metastasis ability. Interestingly, high levels of miR-25 were expressed in both highly metastatic HCC cell-derived exosomes and recipient cells. Mechanistically, miR-25 activates Wnt/β-catenin signaling by reducing the expression level of serine/threonine-protein kinase 1 (SIK1) [78]. 

Yu et al. analyzed paired tumor tissues and adjacent non-tumor tissues of HCC patients, including 35 paraffin-embedded and 40 frozen specimens. They found that DEAD-box helicase 55 (DDX55) was overexpressed in HCC tissues, mainly located in the nucleus and cytoplasm. They studied its function and mechanism and found that DDX55 interacted with Bromodomain-containing protein 4 (BRD4) to form a transcriptional regulatory complex that positively regulated the transcription of PIK3CA, a core gene regulating Akt and Wnt pathways. Subsequently, the PI3K/Akt/GSK-3β pathway was activated, stabilizing β-catenin and regulating the downstream gene expression, ultimately promoting cell cycle progression and EMT. On further investigation, it was noted that DDX55 could be transferred from HCC cells with relatively high DDX55 expression to those with relatively low DDX55 expression via exosomes. They also observed that these DDX55-rich exosomes could also be transferred into endothelial cells, inducing increased angiogenesis, and the underlying mechanism might be related to the activation of the β-catenin signal [79].

In one study, serum samples were collected from randomized healthy donors, patients with chronic HBV infection, and patients with liver disease (cirrhosis, early HCC, and late HCC) without any treatment. Then exosomes were extracted from the serum. Interestingly, the researchers found that patients with late HCC had the greatest number of serum exosomes. Then they co-cultured these exosomes with Huh7 cell lines and were surprised to find that exosomes from HCC patients increased the growth and motility of Huh7 cells. In addition, exosomes derived from patients with late HCC were superior to those from cirrhotic patients and even more superior than those from patients with early HCC. This finding was linked to the promotion of malignant biological behavior of HCC cell lines by exosomes in vitro or the promotion of tumor formation and metastasis in orthotopic liver transplantation models in vivo. The research focused on the underlying mechanism and found that the polymeric immunoglobulin receptor (pIgR) was abundant in exosomes derived from advanced HCC patients. These exosomes were primarily secreted by tumor cells. When recipient HCC cells took up these exosomes, β-catenin was transferred from the cytoplasm to the nucleus, which was closely related to the activation of PDK1/Akt/GSK-3β signaling. These exosomes ultimately enabled the recipient cells to acquire stem cell properties and enhance their invasion and metastasis abilities [80].

Among the many characteristics of cancers, abnormal sialylation has often been found to be important for the progression of tumors [81]. Wang et al. discovered that when α2,6-sialyltransferase I (ST6Gal-I) was silenced, the expression level of CD63, a regulator of exosome production and secretion, decreased. The level of α2,6-sialylated glycoconjugates on the surface of HCC-derived exosomes also decreased, reducing the internalization of exosomes by recipient cells. Furthermore, silencing of this molecule could attenuate the highly metastatic attributes of HCC-derived exosomes on the malignant biological behavior of low metastatic HCC cells, related to the suppressed function of highly metastatic HCC-derived exosomes, resulting in the failure of Akt/GSK-3β/β-catenin signaling and JNK1/2 signaling [82].

**Table 1 cancers-15-02088-t001:** EVs derived from HCC cells.

Molecules	Origin	Recipient Cells	Effects on Cell Behaviors	Mechanisms	References
MiR-1273f	Hypoxic HCC cells	Normoxic HCC cells	Promote proliferation, migration, invasion, and EMT	Upregulate LHX6/β-catenin	[70]
MiR-25	High-metastatic HCC cells	Low-metastatic HCC cells	Reduce apoptosis and promote proliferation, migration, and invasion	Upregulate SIK1/β-catenin	[78]
DDX55	HCC cells	HCC cells	Promote cell cycle progression and EMT.	Upregulate PI3K/Akt/GSK-3β/β-catenin	[79]
		Endothelial cells	Promote angiogenesis	Upregulate PI3K/Akt/GSK-3β/β-catenin	[79]
PIgR	Advanced HCC cells	Early HCC cells	Acquire stem cell properties and promote migration and invasion	Upregulate PDK1/Akt/GSK-3β/β-catenin	[80]
Not mentioned	High-metastatic HCC cells	Low-metastatic HCC cells	Promote proliferation and migration	Upregulate Akt/GSK-3β/β-catenin and JNK1/2 signaling	[82]

### 2.2. EVs Modulate Wnt Signaling through Transport between HCC and Stromal Cells and between Different Types of Stromal Cells

In addition to transferring cargos between tumor cells, EVs also act as a bridge between tumor cells and stromal cells and even between different types of stromal cells (Table 2). CAFs are important components of TME and are important for cancer progression and resisting treatment. Qin et al. discovered that Gremlin 1 was highly expressed in CAF-derived exosomes, which could be transferred into HCC cells to promote invasion, migration, and EMT of the recipient cells by activating Wnt/β-catenin signaling. As EMT is closely related to sorafenib resistance, they also investigated the effect of exosome-mediated Gremlin 1 transfer on the sensitivity of HCC cells to sorafenib. Their results indicated that HCC cells were less sensitive to sorafenib after exosome treatment through a similar mechanism, as described above [83]. Additionally, another study found that the expression of miR-320a notably decreased in HCC tissues and liver cancer cell lines. Lu et al. found that miR-320a overexpression in the HepG2 cells exhibited a marked inhibitory effect on cell proliferation. Mechanically, miR-320a exerted its tumor-suppressive function via downregulating the Wnt/β-catenin signaling pathway. However, in this study, for the first time, miR-320a was reported as a negative regulator of β-catenin [84]. Additionally, Zhang et al. found a significant reduction in the miR-320a expression in CAF-derived exosomes. CAF-mediated HCC tumor progression is partially related to the loss of miR-320a in the exosomes of CAFs, and transferring stromal cell-derived miR-320a might be a potential treatment option against HCC progression [85].

Macrophages are an important part of immune cells and mainly include two subtypes, namely: M1 type and M2 type. Among them, the M1 type plays a tumor suppressor role, whereas the M2 type has tumor-promoting properties, TAMs, with the M2 phenotype, in many tumors associated with malignant tumor progression [86,87]. Liu et al. found that miR-92a-2-5p is highly expressed in M2 macrophages and can be secreted into exosomes. When the exosomes were used to treat HCC cells, they found that the invasive ability of the recipient cells was enhanced. Then they explored the mechanism and observed reduced androgen receptor (AR) expression in recipient cells caused by the inhibition of AR mRNA translation by miR-92a-2-5p. The PHLPP/p-AKT/β-catenin signaling was altered by miR-92a-2-5p/AR axis [88]. Numerous studies have confirmed the association of tumor immune escape with CD8+ T cell dysfunction and severe depletion [89,90,91]. Pu et al. found increased expression of miR-21-5p in M2 macrophage-derived exosomes. After CD8+ T cells were treated with these exosomes, the levels of CD8+ T cell surface exhaustion markers, such as PD-1 and T cell immunoglobulin and mucin domain-containing protein 3, were significantly increased, and that of tumor suppressor factors, such as interferon-γ, interleukin-2, and tumor necrosis factor-α, were reduced. In animal experiments, it was found that exosomes can promote HCC tumorigenesis and malignant behavior and reduce the infiltration of CD8+ T cells in HCC tissues. Mechanistically, miR-21-5p acted mainly by binding to YOD1 mRNA, reducing YOD1 level, and thereby activating YAP/β-catenin signaling [92]. Previous studies had found that the activity of YOD1 was essential for antigen presentation, which was also the key associated with the effective functioning of antigen-specific CD8+ T cells [93]. In other tumor types, YAP has been found to induce the expression and secretion of multiple chemokines, which recruits myeloid-derived suppressor cells (MDSCs) and inhibits the function of cytotoxic T lymphocytes (CTLs), thereby promoting the formation of immunosuppressive TME [94,95].

**Table 2 cancers-15-02088-t002:** EVs derived from CAFs and M2 macrophages.

Molecules	Origin	Recipient Cells	Effects on Cell Behaviors	Mechanisms	References
Gremlin 1	CAFs	HCC cells	Promote migration, invasion, and EMT and reduce sensitivity to sorafenib	Upregulate Wnt/β-catenin	[83]
miR-320a	CAFs	HCC cells	Inhibit proliferation	Downregulate Wnt/β-catenin	[84,85]
MiR-92a-2-5p	M2 macrophages	HCC cells	Promote invasion	Upregulate PHLPP/p-AKT/β-catenin	[88]
MiR-21-5p	M2 macrophages	CD8+ T cells	Induce exhaustion and reduce intratumoral immune infiltration	Upregulate YAP/β-catenin	[92]

### 2.3. EVs Modulate β-Catenin Signaling in HCC by Altering the Cellular Localization of β-Catenin

Previous studies showed that the cellular localization of β-catenin was closely related to its function. When accumulated in the nucleus, it promoted EMT by mediating the expression of target genes. However, when localized on the plasma membrane (PM), it interacted with E-cadherin to increase cell-to-cell adhesion [96,97]. Several studies have found that EVs alter the cellular localization of β-catenin, which modulates β-catenin signaling in HCC (Table 3). Han et al. found that Vps4A overexpression could mediate β-catenin relocalization, promoting β-catenin expression on PM and secreting it out of the cell through exosomes, thereby reducing nuclear β-catenin level, which downregulated β-catenin signaling and suppressed EMT [98]. Then the research group found a similar phenomenon on another molecule. They found that exosomes derived from p62-overexpressing HCC cells can be taken up by p62-low-expressing HCC cells, following which p62 increased intracellular GSK3β levels, finally promoting β-catenin degradation. Unexpectedly, they observed that high levels of p62 promoted the malignant phenotype of HCC, including cell proliferation, migration, and invasion. This may be related to the p62-mediated extracellular secretion of exosome-coated β-catenin [99].

## 3. EVs-Based Biological Nanoparticles Modulate Wnt Signaling in HCC Cells

Desirable intrinsic properties of EVs, such as the ability to bypass natural membranous barriers and to deliver their unique biomolecular cargo to specific cell populations, position them as fiercely competitive alternatives for currently available cell therapies and artificial drug delivery platforms [100]. EVs-based biological nanoparticles, an emerging therapeutic delivery system, are a promising therapeutic strategy due to their small size, strong tissue penetration, high stability in circulation, inherent cell-targeting capacity, and ability to overcome innate immune barriers [101]. As β-catenin activation has a role in the occurrence and development of HCC, the use of EVs-based biological nanoparticles loaded with β-catenin siRNA for the HCC treatment can be a feasible solution. Matsuda et al. found that therapeutic milk-derived nanovesicles (tMNVs) loaded with β-catenin siRNA could target β-catenin in HCC in vivo and in vitro and inhibited β-catenin activity, thereby slowing tumor growth and increasing the therapeutic effect of immune checkpoint inhibitors [102].

A growing body of research has found that tumor-initiating cells, such as liver cancer stem cells (LCSCs), contribute to tumor growth and resistance to therapy, which has led to increased interest in these cells. This provides the idea for improving the therapeutic response of HCC by specifically targeting this cell population. LCSCs have various molecular markers, among which epithelial cell adhesion molecule (EpCAM) is a marker of cancer-initiating cells in the liver and other epithelial tissues. EpCAM expression is regulated by Wnt/β-catenin signaling and controls hepatic stem cell proliferation [103,104,105]. Ishiguro et al. developed an engineered EV that is based on tMNV. It specifically targets EpCAM-expressing LCSCs in vivo and in vitro and delivers β-catenin siRNA to recipient cells, thereby inhibiting β-catenin activity and tumor proliferative [106].

MiR-375 is a tumor suppressor microRNA, and its expression is downregulated in various tumors, such as colorectal cancer, pancreatic cancer, and breast cancer. Its high expression can suppress the malignant phenotype of tumors by directly binding to HOXB3 [107,108,109]. In a study, the bone marrow-derived mesenchymal stem cells (BM-MSCs) were transfected with miR-375, and the molecule was successfully expressed in BM-MSCs and exosomes. Subsequently, HCC cells were treated with these isolated miR-375-enriched exosomes. The malignant behaviors of HCC cells, including proliferation, migration, and invasion, were inhibited. Mechanistically, miR-375 exerted its action by regulating the HOXB3/Wnt/β-Catenin axis after entering HCC cells [110]. Owing to the advantages of the wide distribution of and easy access to Human BM-MSCs (h-BMMSCs), their great potential in self-renewal and multi-lineage differentiation of h-BMMSCs, and their ability to avoid immune rejection during autologous transplantation, h-BMMSCs show potential for wide clinical application [111,112].

## 4. Conclusions and Future Directions

HCC is undoubtedly the most common type of liver cancer, and as a highly malignant tumor, it has a high mortality every year. Unfortunately, patients often miss timely treatment because of the difficulty of its early detection. EVs have been a hot topic of research in recent years. Owing to their unique structure and material loading capacity and their ability to mediate intercellular communication, EVs play a pivotal role in the occurrence and development of various diseases, including cancers. They, therefore, have great potential for cancer treatment. Wnt signaling is involved in many physiological and pathological processes and can be mainly divided into canonical and noncanonical pathways. Aberrant regulation of Wnt signaling has been observed in the development and progression of HCC, along with the involvement of EVs. Therefore, summarizing the recent research progress of EVs regulating Wnt signaling in HCC can give us a better understanding of this disease and may provide a way to break the deadlock of HCC treatment.

In recent studies, the regulation of Wnt signaling in HCC by EVs mainly focused on regulating the canonical pathway, whereas studies on the noncanonical pathway are still lacking. EVs are involved in the regulation of canonical pathway in various ways, such as mediating intercellular transport between different types of HCC cells and between HCC and stromal cells. Interestingly, EVs are found to modulate Wnt signaling by mediating the cellular localization of β-catenin. In addition, EVs-based biological nanoparticles also inhibited Wnt signaling by delivering inhibitory molecules such as β-catenin siRNA into HCC cells, thereby inhibiting tumor progression. It is not difficult for us to find that EVs act as carriers for cargo delivery and regulate the biological behavior of the recipient cell in HCC TME (Figure 2). Based on the aforementioned constructive findings, future research directions can be considered from the following aspects: 1. Since TME also contains a large number of immune cells, HCC immunotherapy is bound to be a hot topic. At present, the regulation of Wnt signaling by EVs is mostly focused on other types of stromal cells, and research on immune cells is still lacking. Therefore, exploration in this field will be very valuable; 2. Currently, there is still a lack of research on the Wnt noncanonical pathway in HCC. Considering that it is also an important part of Wnt signaling and it has been studied in other tumors, the study on this aspect will be of great significance. We believe that with more research in this field, how EVs regulate Wnt signaling in HCC will become clearer, which will certainly help us to form a regulatory network and bring us some inspiration. It will open up a sunny avenue for the treatment of HCC.

## Figures and Tables

**Figure 1 cancers-15-02088-f001:**
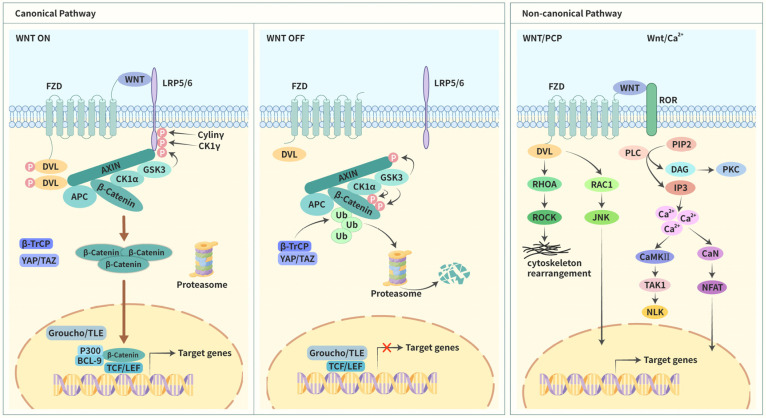
Schematic interpretation of canonical and noncanonical Wnt pathway.

**Figure 2 cancers-15-02088-f002:**
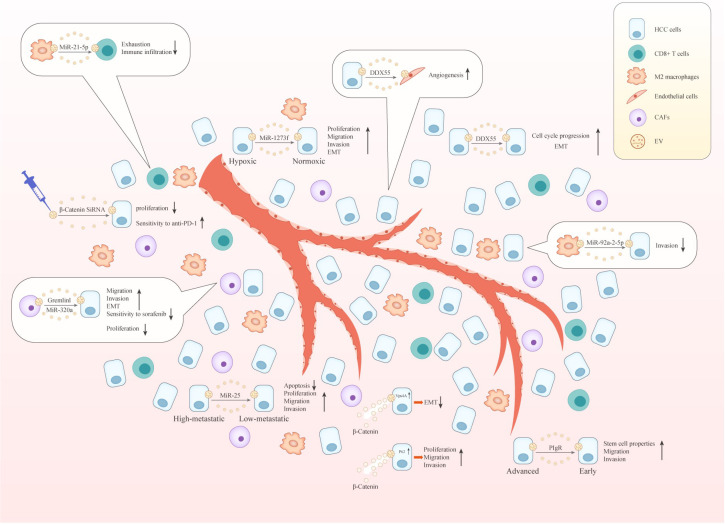
EVs act as cargo delivery carriers and regulate the recipient cell’s biological behavior in HCC TME. ↑: promotion; ↓: inhibition.

**Table 3 cancers-15-02088-t003:** EVs alter the cellular localization of β-catenin.

Molecules	Origin	Recipient Cells	Effects on Cell Behaviors	Mechanisms	References
Vps4A			Reduce EMT	Regulate cellular localization of β-catenin	[98]
p62			Promote proliferation, migration, and invasion	Regulate cellular localization of β-catenin	[99]

## Data Availability

The data can be shared up on request.

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
