# Peer review of "Extracellular Vesicles Act as Carriers for Cargo Delivery and Regulate Wnt Signaling in the Hepatocellular Carcinoma Tumor Microenvironment"

_cancers, 2023, doi:10.3390/cancers15072088_

Round 1

Reviewer 1 Report (Previous Reviewer 1)

The Authors have addressed all of my concerns. 

Author Response

So much thanks for your affirmation and approval to our manuscript.

Reviewer 2 Report (New Reviewer)

The most prevalent and deadliest form of cancer is hepatocellular carcinoma (HCC), which also has a high mortality rate. The current course of treatment comprises early detection and surgical tumor removal. Few therapeutic options are available, most of which are localized and mostly consist of chemotherapy, targeted therapy, radiofrequency ablation (RFA), and transarterial chemoembolization (TACE). Unfortunately, the treatment is typically inadequate and adversely affects patients' chances of survival. This article summarizes the current literature on regulating Wnt signaling by extracellular vesicles (EVs) in HCC and their underlying mechanisms. EVs are well known for carrying a specific composition of proteins, lipids, and nucleotides and have been considered essential mediators of cell-cell communication. EVs participate in developing diseases, including HCC, by regulating cell signaling pathways. Recently, a number of research and review articles on exosome-mediated signaling were published. I noticed no extra information for the readers in this article. Even though the figures and tables are instructive, I still have a few more concerns.

The authors should include the details of the clinical studies.

When describing Wnt signaling regulation in HCC, the authors should describe the various exosome types separately.

How do chemokines and exosomes interact to regulate HCC? In addition, drug resistance caused by exosomes, immunotherapy, and Wnt-induced cellular metabolism should all be considered.

Author Response

1.The authors should include the details of the clinical studies.

Response to reviewer: We appreciate this valuable suggestion, this will help refine our manuscript. Therefore, we add the details of the clinical studies in our manuscript.

Revisions: “Yu et al. analyzed paired tumor tissues and adjacent non-tumor tissues of HCC patients, including 35 paraffin-embedded specimens and 40 frozen specimens, and found that DEAD-box helicase 55 (DDX55) was overexpressed in HCC tissues and mainly located in the nucleus and cytoplasm. They studied its function and mechanism and found that DDX55 interacted with Bromodomain-containing protein 4 (BRD4) to form a transcriptional regulatory complex that positively regulated the transcription of PIK3CA, a core gene regulating Akt and Wnt pathways. Subsequently, the PI3K/Akt/GSK-3β pathway was activated, stabilizing β-catenin and regulating the downstream gene expression, ultimately promoting cell cycle progression and EMT”(2.1. section, page 6)

“In one study, serum samples were collected from randomized healthy donors, patients with chronic HBV infection, and patients with liver disease (cirrhosis, early HCC, and late HCC) without any treatment, and then exosomes were extracted from the serum. Interestingly, the researchers found that patients with late HCC had the greatest amount of serum exosomes. Then they co-cultured these exosomes with Huh7 cell lines, they were surprised to find that exosomes from HCC patients increased the growth and motility of Huh7 cells.”(2.1. section, page 6-7)

2.When describing Wnt signaling regulation in HCC, the authors should describe the various exosome types separately.

Response to reviewer: Thank you for your valuable opinions. Therefore, we carefully review the diameters, morphology and biomarkers of EVs in the articles and distinguish different types of exosomes when we describe Wnt signaling regulation in HCC.

3.How do chemokines and exosomes interact to regulate HCC? In addition, drug resistance caused by exosomes, immunotherapy, and Wnt-induced cellular metabolism should all be considered.

Response to reviewer: Thank you for your valuable and thoughtful comment. We looked further into the articles and added to this section.

Revisions: “Current studies have found that EVs can transfer contents between different HCC cells and activate Wnt signaling, thereby regulating the biological behavior of recipient cells and promoting the progression of malignant phenotypes. In addition, stromal cell-derived EVs can also deliver contents to HCC cells and other stromal cell types, these, on the one hand, lead to HCC resistance, on the other hand, lead to immune cells expressing more exhaustion markers and fewer effector molecules. ”(2. section, page 5-6)

“In other tumor types, YAP has been found to induce the expression and secretion of multiple chemokines, which recruits myeloid-derived suppressor cells (MDSCs) and inhibits the function of cytotoxic T lymphocytes (CTLs), thereby promoting the formation of immunosuppressive TME.”(2.2. section, page 8)

Reviewer 3 Report (New Reviewer)

This is a well put together` review.My main suggestions are to improve the layout to allow more easy readings .

Hence I would suggest

1.Breaking up Table 1 into 3 separate tables scattered throughout the paper adjacent to each relevant section

e.g.

Table 1 -EVs derived from HCC cells

Table 2 -EVs derived from CAFS

Table 3 EVs derived from M2 macrophages

2.Breakinh up sections 2.1 , 2.2, 3.into  a few paragraphs per section

3.List your research recommendations at the end rather then set out as they are.

Author Response

1.Breaking up Table 1 into 3 separate tables scattered throughout the paper adjacent to each relevant section

e.g.

Table 1 -EVs derived from HCC cells

Table 2 -EVs derived from CAFS

Table 3 EVs derived from M2 macrophages

Response to reviewer: Thanks for your advice. We break up Table 1 into 3 separate tables. Since CAFs and M2 macrophages are in the same part, they are put together into Table 2. And Table 3 is adjacent to 2.3. section.

Revisions: 

“Table 1. EVs derived from HCC cells”

“Table 2. EVs derived from CAFs and M2 macrophages”

“Table 3. EVs alter the cellular localization of β-catenin”

2.Breaking up sections 2.1 , 2.2, 3.into  a few paragraphs per section

Response to reviewer: Thank you for reminding us of this important issue. We break up sections 2.1 , 2.2, 3.into a few paragraphs per section.

3.List your research recommendations at the end rather then set out as they are.

Response to reviewer: Thank you very much for your conscientious comment on our manuscript. We revise the last part and list our research recommendations.

Revisions: “Based on the aforementioned constructive findings, future research directions can be considered from the following aspects: 1. since TME also contains a large number of immune cells, and the HCC immunotherapy is bound to be a hot topic, at present, the regulation of Wnt signaling by EVs is mostly focused on other types of stromal cells, and the research on immune cells is still lacking, therefore, the exploration in this field will be very valuable; 2. currently, there is still a lack of research on the Wnt noncanonical pathway in HCC, considering that it is also an important part of wnt signaling and it has been studied in other tumors, the study on this aspect will be of great significance.”(4. section, page 11-12)

Round 2

Reviewer 2 Report (New Reviewer)

Every question asked has been answered by the authors. The article might be accepted for publication in this journal.

This manuscript is a resubmission of an earlier submission. The following is a list of the peer review reports and author responses from that submission.

Round 1

Reviewer 1 Report

In the manuscript, the authors attempted to evaluate the role of EVs as modulators of Wnt signaling in HCC. The topic is interesting and worth studying; however, the manuscript is difficult to follow and requires significant revision.

1. Title suggests original work; please change it.

2. The manuscript's text is somewhat conversational, and a native speaker must modify the English language.

3. All abbreviations should be explained.

4. The description of EVs' role is unclear - different concepts are given one after the other without logical order.

5. It is difficult to draw unambiguous conclusions from table 1 - without analyzing the references, the reader is not able to draw any constructive findings from it,

6. The authors did not discuss the critical works on this subject, namely:

doi: 10.3390/cancers12103019

doi: 10.1111/liv.14585doi: 10.3390/cancers13123076

doi: 10.3390/cells11030490

doi: 10.3390/cells11131989 

doi: 10.1136/gutjnl-2021-325036

Reviewer 2 Report

Comments and criticisms

The authors summarized the recent reports of EVs associated with Wnt signaling pathway in hepatocellular carcinoma. In general, this review manuscript well covers the EVs, but it lacks several important issues and contains flaws and errors. Therefore, the following points should be revised.

(Major point)

The manuscript does not overview the tumor microenvironment (TME) containing cancer cells and stromal cells. Therefore, inclusion of a section overviewing TME is essential.

Table 1 lacks exosomal miR-320a derived from CAF in HCC tissues.

Since GOLM is not involved in the Wnt signaling pathway (ref[79]), it should be deleted from Table 1.

Inclusion of beta-catenin siRNA (ref[94], [98]) in Table 1 is not appropriate, because synthetic small interfering RNA (siRNA) was delivered within EVs as a therapeutic strategy.

Inclusion of a cartoon depicting different interactions by EVs between cancer cell and cancer cells, cancer cells and stromal cells, and cancer cells and immune cells is recommended.

Section 2.4 should be changed to section 3 (and section 3 to section 4) because this section explains therapeutic strategy using EVs.

 (Minor issues)

Abbreviations should be spell out when they appeared first time in this manuscript, e.g. RFA and TACE in p2 (introduction).

The following description include poor English writing, “it is necessary and urgent to study the mechanism of its occurrence and development and the latest research progress, these will help us learn more about the disease and help us find new and effective treatments”. It should be revised.

In addition, I found a lot of grammatical errors. Therefore, this manuscript should be carefully checked by a native English speaker(s) and the writing should be revised throughout the manuscript.